# Current Societal Views about Sustainable Wildlife Management and Conservation: A Survey of College Students in China

**DOI:** 10.3390/ani10101821

**Published:** 2020-10-06

**Authors:** Zhen Miao, Qiang Wang, Dongxiao Chen, Zhifan Song, Wei Zhang, Xuehong Zhou, Douglas C. MacMillan

**Affiliations:** 1College of Wildlife and Protected Area, Northeast Forestry University, Harbin 150040, China; miaozhen43566@163.com (Z.M.); dongxiao_c@163.com (D.C.); 13136751029@163.com (Z.S.); 2Key Laboratory of Wetland Ecology and Environment, Northeast Institute of Geography and Agroecology, Chinese Academy of Sciences, Changchun 130102, China; qwang@neigae.ac.cn; 3Durrell Institute of Conservation and Ecology (DICE), University of Kent, Canterbury, Kent CT2 7NR, UK; dcm@kent.ac.uk

**Keywords:** wildlife conservation, sustainable wildlife management, animal welfare and rights, attitude, China, college students

## Abstract

**Simple Summary:**

Wildlife conservation and management has become a very complex public policy issue in China as concerns about animal welfare and empathy for animals have emerged, especially in the younger generation. Science-based conservation policy and strategy that focus on sustainable management are perceived as less irrelevant today and can often be in conflict with emerging attitudes and values. Sustainable wildlife management adheres to the conservation concept of effective combination of species conservation and sustainable utilization, with an aim to establish a long-acting species conservation model that promotes all-round development of ecology, society, and economy, and has traditionally formed the basis of conservation management. This study designed a semi-structured questionnaire, which aimed to assess attitudes of Chinese college students towards sustainable wildlife management and wildlife conservation, and investigate the role of demographic and other characteristics on attitude. From October 2018 to April 2019, nine universities (including “Double First-Class” universities, first-tier universities, second-tier universities), and four three-year colleges in China were selected as survey sites, where face-to-face interviews were conducted among students. The results show that students broadly support the sustainable wildlife management but not in issues relating to “Animal Welfare and Rights” and “Trophy Hunting”. Students with lowest support for the theory and practice of sustainable wildlife management are vegetarians, freshmen, and those who have taken environmental protection electives in their educational program.

**Abstract:**

Wildlife conservation and management has become a very complex public policy issue in China as concerns over on animal welfare and empathy for animals have grown. Science-based conservation strategies that are oriented toward sustainable wildlife management (SWM) are under threat as these new attitudes and values emerge and take hold. This study accesses the attitudes of college students towards SWM and wildlife conservation, and investigates demographic characteristics influencing their attitudes in China, a country that is traditionally associated with consumptive use of wildlife and SWM, but where new ideas about wildlife conservation are emerging. From October 2018 to April 2019, nine universities (including “Double First-Class” universities, first-tier universities, second-tier universities), and four three-year colleges in China were selected as survey locations, and face-to-face interviews were conducted with 1991 students. A total of 1977 questionnaires were recovered, of which 1739 were valid, with a completion rate of 88%. A Likert seven-point scale method was used to score students’ attitudes, and a classification and regression tree (CART) was used to analyze whether their attitudes were affected by their demographic characteristics. The results show that although students are broadly supportive of the theory of SWM, some are deeply antagonistic about on SWM on issues that arouse strong emotions such as “Animal Welfare and Rights” and “Trophy Hunting”. Demographic characteristics of students affect their degree of support for the SWM with support for SWM lower among vegetarians, freshmen, and students who have taken environmental protection electives. This research suggests that the theory of SWM requires to be refreshed and adapted to appeal to the younger generation of Chinese students, with SWM principles integrated into the environmental education programs of universities and three-year colleges. More attention should also be attached to media publicity by the government about wildlife conservation so as to enhance awareness of the need for SWM.

## 1. Introduction

Across the world, fundamental changes to human–animal relationships are taking place. The rapid change in attitudes and values towards animals and the increasing attention to the compassionate conservation are raising profound questions for sustainable wildlife management and the conservation of species [1]. The use of wildlife for scientific laboratory research [2], the wildlife trade for meat [3], Chinese medicine [4], lethal control of invasive species [5], captive breeding of wildlife [6], and farming for fur are now deeply contentious and the preeminent role of science is being called into question in a world where ‘feelings’ may matter as much as ‘facts’ [7]. As emphasized by Hampton [8], arguments that arouse compassion can often trump scientific knowledge, with policy and management increasingly being pushed on to a more subjective footing.

In China, attitudes and values affecting conservation are also changing quickly as the Chinese government has introduced a range of policies and special measures to promotion of awareness about nature with activities, such as “Wildlife Conservation Month” and “Bird Loving Week” [9], and by the introduction of laws to reduce the indiscriminate consumption of wildlife, such as a ban on ivory markets and the consumption of key protected wildlife species such as common Leopard (*Panthera pardus*) and all species of Moschidae. Furthermore, the outbreak of the Corona Virus Disease 2019 (COVID-19) has further affected the human-animal relationships in Chinese society. Although the origin of the COVID-19 has not yet been determined, it has caused a dramatic change of attitudes of public and government towards wildlife conservation and utilization [10]. In order to promote wildlife conservation and prevent possible pandemics in the future, China enacted laws emphasizing the prohibition of consumption of all wildlife, and imposed more severe penalties for violations of the wildlife consumption ban [10,11].

Wildlife conservation and management is therefore becoming a very complex and challenging public policy issue in China. The theory and management tools of traditional sustainable wildlife management (SWM) are under threat as new attitudes and values for wildlife emerge. SWM adheres to the conservation concept of effective combination of conservation and sustainable utilization [12,13], with an aim to establish a long-acting species conservation model that promotes all-round development of ecology, society, and economy [14]. SWM has traditionally formed the basis of conservation management and is promoted by major international conservation organizations, including Convention on International Trade in Endangered Species of Wild Fauna and Flora (CITES), Convention on Biological Diversity (CBD), International Council for Game and Wildlife Conservation (CIC), and International Union for Conservation of Nature (IUCN) [15]. However, emotional attachment to individual animals, and concerns about individual animal welfare in general are deepening in intensity, and may supersede this science-based conservation thinking and strategies especially among the young generation by promoting compassion and love above other considerations. For example, there has been a rapid increase in the number of ‘good-hearted’ people who purchase wildlife at the market and release them into the environment [16], despite scientific evidence that the animals do not survive and may threaten extant populations by introducing disease [17].

Conservation scientists need to gain a better understanding of these diverging attitudes and values in order to build consensus and find new solutions for wildlife conservation in China and elsewhere. In this study we seek to understand the reasons why people make certain decisions or exhibit certain behaviors using an attitude survey of young Chinese students, where newer ideas about the wildlife–human nexus are emerging. Attitude studies can help researchers understand people’s concepts and values on wildlife conservation, assist managers to formulate more scientific and comprehensive wildlife conservation strategies that also reflect public awareness and concerns about wildlife-related issues [18,19].

Our key aim is to study the attitudes of college students to the theory of SWM and wildlife conservation in the context of rising support for animal welfare to identify specific areas of conflict between SWM and emerging attitudes of China’s young generation, and further investigate whether their demographics will affect their attitudes. We focus on college students because (i) they are more open-minded when accepting animal welfare, which is still an emerging phenomenon in China [20]; (ii) they have more educational opportunities related to animal welfare and, therefore, are considered to be more concerned about animal welfare than the older generation [9,20]; and (iii) they will be the ‘influencers’ and leaders in China, and their views about sustainable development and future wildlife management will be key to how SWM evolves in the future [21]. Specifically we explore the hypothesis that: college students as a whole are broadly supportive of the SWM (hypothesis 1); demographic characteristics of college students (e.g., vegetarian habits, grades, gender, major, and interest in wildlife-related activities) can affect their degree of support for the SWM (hypothesis 2).

## 2. Materials and Methods

### 2.1. Data Collection

Data collection was carried out between October 2018 and April 2019 using a stratified sampling approach and a semi-structured questionnaire at nine universities and four three-year colleges in China. To reflect the different educational levels in the Chinese system, with different educational resources, teaching qualifications, and student academic ability, we divided the institutions into four strata (“Double First-Class” university, first-tier university, second-tier university, and three-year college) with 13 randomly selected for face to face questionnaires with students (Table 1). A total of 1991 questionnaires were distributed and 1977 questionnaires were recovered in this survey. After descriptive statistical preliminary analysis, incomplete questionnaires were eliminated; there was 1739 valid questionnaires with a completion rate of 88%.

### 2.2. Questionnaire Design

The questionnaire (see Appendix B) consisted of two parts. In the first part, we ascertained the demographic characteristics of the respondents, including gender, grade, major (agriculture, science, engineering, medicine, economics, management, law, literature, fine arts), and recorded whether the respondent was a vegetarian (abbreviation: vegetarian), whether the respondent had ever paid attention to information related to wildlife conservation (abbreviation: attention), whether the respondent taken environmental protection electives during university/three-year college (abbreviation: electives) or participated in wildlife related activities (abbreviation: activity).

In the second part, we designed a series of questions to analyze beliefs and attitudes towards the theory of SWM and wildlife conservation with topics covered ranging from animal welfare [9], captive breeding of wildlife [22], wildlife release [16], and vegetarianism [23]. We adapted Fulton’s (1996) attitudes scales [24,25,26,27], using 20 questions from both forward (questions 4, 5, 6, 8, 9, 12, 13, 16, 19, 20) and reverse (questions 1, 2, 3, 7, 10, 11, 14, 15, 17, 18) angles. For analysis, we grouped the 20 questions into seven main categories of contemporary relevance to the SWM debate:(a)Release: Question 1 (Q1): people should be able to release wild animals at will; Question 13 (Q13): wildlife should be released by professional departments or organizations; Question 15 (Q15): buying wildlife from the market and releasing them is beneficial to conservation;(b)Animal Welfare and Rights: Question 2 (Q2): animals should have equal rights with humans; Question 10 (Q10): advocating animal rights is more important than utilization of wildlife by men; Question 16 (Q16): animal welfare should be improved in captive breeding of wildlife;(c)Utilization and Wildlife Conservation: Question 3 (Q3): as long as the use of wild animals and their products is prohibited, wild animals can be effectively conserved; Question 18 (Q18): all wild animals and their products, including captive breeding of wild animals and their products, should not be used.(d)Wildlife Management: Question 4 (Q4): if wild populations are not threatened, we can use wildlife and their products to improve people’s quality of life; Question 8 (Q8): the standard of living of residents around the wildlife distribution area should be considered in wildlife conservation; Question 20 (Q20): wildlife management should be based on science;(e)Vegetarianism and Wildlife Conservation: Question 5 (Q5): vegetarianism is not an effective way to conserve and manage wildlife; Question 11 (Q11): vegetarianism is closely related to wildlife conservation; Question 17 (Q17): wild animals can be effectively conserved if all the human beings are vegans;(f)Public and Wildlife Conservation: Question 6 (Q6): if all men have a positive attitude towards wildlife, wildlife can be effectively conserved; Question 14 (Q14): wildlife can be conserved as long as given care and love; Question 19 (Q19): no matter how enthusiastic we care about wildlife, we still need scientific methods for its conservation;(g)Trophy Hunting: Question 7 (Q7): trophy hunting is cruel and inhumane to animals; Question 9 (Q9): well-managed trophy hunting is one of the effective measures for wildlife conservation and management; Question 12 (Q12): well-managed hunting opportunities should be provided for those who want to hunt.

Before the official survey, we conducted a preliminary survey in Harbin in October 2018 and analyzed the validity and reliability of the questionnaire. A total of 43 questionnaires were distributed in the pre-survey, and 39 valid questionnaires were recovered, with a completion rate of 91%. We used Cronbach’s Alpha to analyze the reliability of the questionnaire, with the reliability of this questionnaire being 0.701 [28]. We used Kaiser-Meyer-Olkin (KMO) and Bartlett sphere test (KMO and Bartlett sphere test are measures of how suited the data is for factor analysis) for validity, in which the KMO measure of sampling adequacy was 0.743 and the Bartlett spherical test was 322.421, *p* = 0.00 < 0.05 [28].

A Likert seven-point scale was used to assign different scores to each option of the question, and assign positive value to questions in forward angle: strongly disagree = −3, disagree = −2, generally disagree = −1, indifferent = 0, generally agree = +1, agree = +2, strongly agree = +3. Questions in reverse angle were scored in opposite fashion. In order to avoid mode effects, the 20 questions in the questionnaire were sorted randomly [29]. After the survey, the average value of all questions were calculated. The average score obtained represents the respondent’s attitude towards SWM and wildlife conservation. The higher the score is, the more support the respondents’ attitudes and cognitions of related issues with the theory of SWM.

### 2.3. Statistical Analysis

All data was processed and analyzed by SPSS 21.0 (IBM Corporation, Armonk, NY, USA) and Excel (Office version 2016, Microsoft Corporation, Redmond, Washington, WA, USA). Excel was used to count respondents’ demographic characteristics, such as gender, major, grade, vegetarianism, electives, attention to wildlife conservation, participation in activities, etc., and to calculate the scores of seven categories of issues and 20 questions in the questionnaire. The classification and regression tree (CART) was applied to analyze whether the interviewee’s demographic characteristics such as gender, major, grade, and vegetarian habits would affect their attitudes. The method of CART replaced or supplemented a variety of traditional statistical techniques, such as analysis of variance, multiple regression, and logistic regression. With this method, researchers could obtain easy-to-interpret results in various complex data, and determine the most important influencing factors in the study through small sample analysis [18].

### 2.4. Ethical Note

All interviewees and potential candidates were anonymized and gave their oral informed consent for inclusion before they participated in the study, and the investigators assured them that all of the information provided would be kept strictly confidential to ensure their privacy. The study was conducted in accordance with the Declaration of Helsinki, and the study was approved by the Ethics Committee of Northeast Forestry University (Project identification code: 2020006).

## 3. Results

### 3.1. Demographics

The results showed that the ratio of male to female respondents was balanced, at 50.9% and 49.1%, respectively. Most of the respondents were freshmen (47.6%), with sophomores, juniors, and seniors accounting for 20%, 17.5% and 6.6%, respectively. Graduate students accounted for 8.3%. Most students majored in science (78.6%), with 21.4% majoring in liberal arts. Among the respondents, 77.4% were non-vegetarians, and 34.4% of the respondents had taken electives related to environmental protection during university/three-year college. Moreover, 28.2% of the respondents had participated in wildlife-related activities, and 39.2% had never been directly paid attention to the wildlife conservation. Students mainly obtained knowledge about wildlife through television (54.2%) and the Internet (60.1%); 33.9% and 30.9% of them learned about information relevant to wildlife through newspapers and schools; fewer students resorted to radio (16.6%), family members (10.8%), friends (14.2%), or other (12%) channels to obtain information related wildlife (see Appendix A in the Appendix A).

### 3.2. Attitude of Students toward SWM and Wildlife Conservation

The total average attitude score with the mean scores for each of the seven categories of issues (Figure 1) calculated for further analysis. The results showed that the total average score of students was 0.503, and their degree of support for seven categories of issues clearly varied. Students scored positive on the issues of Release (1.14), Utilization and Wildlife Conservation (0.19), Wildlife Management (0.89), Vegetarianism and Wildlife Conservation (0.61), and Public and Wildlife Conservation (1.1), among which, the highest score was on Release. By contrast, negative average scores were obtained for Animal Welfare and Rights (−0.28) and Trophy Hunting (−0.24).

The mean scores of 20 questions (Figure 2) were also calculated. The results showed that scores of most of the questions showed a fairly consistent trend, indicating that the respondents’ understanding of most of the questions was consistent with the theory of sustainable wildlife management (SWM). Among the 20 questions, Q19 (that wildlife conservation requires scientific methods (1.813)) and Q20 (that wildlife management should be based on science (1.866)) had the highest scores. Scores of Q2 (−0.89) and Q7 (−1.07) were the lowest, indicating that most respondents supported the view that animals were equal to humans and that trophy hunting was inhumane.

Considering that the scores of some questions were close to “0” or neutral we performed, frequency statistics to explore the distribution across individual questions (Figure 3). The results showed that some questions with scores close to “0” had broadly similar support ‘for’ and ‘against’ rather than indifferent (no opinion). For example, 42.9% supported and 50.2% opposed question 3 (that wild animals can be effectively conserved as long as the use of wild animals and their products is prohibited). The number of students who supported the prohibition of the use of all wildlife and their products in question 18 accounted for 41.3%, and those who opposed 47.8%. The proportion of students who supported and opposed well-managed trophy hunting as a wildlife management measure in question 9 was 41.8% and 49.3%, respectively.

### 3.3. Demographic Effects

We used the CART model to analyze the total average score and perform post-pruning (Figure 4). The results showed that “vegetarian” was the first branch point in the CART and, therefore, the most important influencing factor. The score of vegetarian students (0.288) was lower than that of non-vegetarian ones (0.566). For non-vegetarian students, “grade” was the influencing factor of the second branch point, and the scores of freshmen (0.487) were lower than those of other grades (0.628). Among freshmen, “electives” was the influencing factor of the third branch point. Freshmen who had taken environmental protection electives had a lower score (0.209) than those who had not taken such electives (0.553). The higher the score, the more the student supported sustainable wildlife management. Overall, factors of vegetarianism, student’s grade, and electives could explain the differences in attitudes of most students. Among them, non-vegetarian senior students had the highest support for sustainable wildlife management.

Subsequently, we conducted the CART analysis and post-pruning on the seven categories of issues (see Appendix A and Appendix A in the Appendix A). The results showed that “vegetarian” was the most important influencing factor (the primary influencing factor for “Release”, “Utilization and Wildlife Conservation”, and “Vegetarianism and Wildlife Conservation”), followed by “grade” (the primary influencing factor for “Wildlife Management”, and “Trophy Hunting”), “electives” (the primary influencing factor for “Public and Wildlife Conservation”) and whether they had paid attention to wildlife conservation (the primary influencing factor for “Animal Welfare and Rights”); these four types of factors also affected different issues as secondary influencing factors. The CART model of each category of issue was different in the hierarchy of influencing factors, so each category was unique to some extent.

On the whole, “vegetarian” was the most important influencing factor. Non-vegetarians were more supportive of SWM than vegetarians. Student’s grade was also an important factor, among which freshmen support for SWM was low. Unexpectedly, students who had taken environmental protection electives had a low level of support for SWM. In addition, “attention” had influence on “Animal Welfare and Rights” and “Trophy Hunting” as primary influencing factor, with students who had paid attention to wildlife conservation having a lower support for SWM. Participation in relevant activities, gender, and major also had a certain influence on some issues as secondary influencing factors. We found that students who had participated in wildlife related activities scored lower on issues of “Release” and “Vegetarianism and Wildlife Conservation”. Girls scored lower on “Wildlife Management” but higher on “Public and Wildlife Conservation”. Students’ majors affected their scores on “Animal Welfare and Rights” and “Utilization and Wildlife Conservation”. However, liberal arts students and science students did not have different stances on these two categories of issues, but we found that students majoring in agriculture had higher scores on these two categories.

## 4. Discussion

This research shows that Chinese college students are broadly supportive toward the theory of sustainable wildlife management (SWM), with an overall average score above zero (mean = 0.503), which is consistent with our hypothesis 1. However, the degree of support for SWM varies considerably depending on demographic characteristics, with vegetarianism, grade, and electives being the main factors that shape the attitudes of students. This supports our second hypothesis 2.

### 4.1. Influence of Demographic Factors

Vegetarianism is the most important factor influencing attitudes, with respect to SWM with vegetarian students having an average score far below non-vegetarian students. Studies of vegetarianism have shown that vegetarians have many motivations to eliminate meat consumption, including physical health, environmental protection, and animal care [30], but moral concern about animals is the primary driver of the vegetarian habit [31,32], and this may explain the opposition to SWM.

Freshmen score lower than other students, indicating that their support for the SWM is weaker than more advanced students. Our results are supported by previous studies, which found that students’ professional and scientific knowledge will be accumulated as they advance in their education and develop a more rational view on the relationship between animal conservation and utilization [33,34]. Perhaps surprisingly, students with most opportunities to take an elective module in environmental protection had significantly less support for SWM. Contrary to our results, previous studies have demonstrated that environmental education can effectively promote students’ understanding of the concept of sustainability and issues related to wildlife conservation [35,36]. Our results suggest that current electives on conservation in Chinese universities and three-year colleges may be less focused on SWM principles than in the past [9] and taken by students who are more interested in animal rights and love of nature. In recent years a number of animal protection alliances have been formed in multiple universities (the purpose of which is to improve college students’ ethical and legal awareness of environmental and animal protection to strengthen their sense of responsibility to the environment) [37], but these initiatives may lack balance regarding SWM. This issue is clearly of great importance and it may therefore be that environmental education has been ‘captured’, or at least overly influenced, by animal rights and freedoms, and lack balance with respect to SWM but further research is merited.

### 4.2. Attitudes towards Seven Categories of Issues

In our study, students are broadly supportive of SWM, but with varying degrees of support for the seven categories (Figure 1). Students are most strongly supportive of contemporary SWM over categories such as “Release”, “Utilization and Wildlife Conservation”, “Wildlife Management”, “Vegetarianism and Wildlife Conservation” and “Public and Wildlife Conservation”, but are less supportive on issues that arouse strong emotions such as “Animal Welfare and Rights” and “Trophy Hunting”.

#### 4.2.1. Animal Welfare and Rights

Animal Welfare and Rights has the lowest mean score overall and is also emerging as a major conflict issue for mainstream conservationists. In our study, 66% of students agreed that animals should have equal status with humans, and 58.9% of students believed that animal rights are more important than human utilization (Figure 3). Similar findings were reported by Davey et al.—that most Chinese college students disagree that humans have the right to use animals when they think fit [38]. While this study did not directly ask about the distinction between animal welfare and animal rights, in the course of the investigation, we also found that some students confused animal welfare with animal rights and think that advocating animal rights or even participating in animal rights movements are to better improve animal welfare, this also mirroring the views of western public [39].

In recent years, with more forums on animal protection organized by animal rights organizations, and the extensive publicity of social media on caring for animals [7,40], Chinese college students’ views on the human–animal relationship have gradually changed, and are paying greater attention to the animal welfare and animal rights. However, due to ambiguities in the social media communication example, animal welfare (humans can reasonably use or scientifically manage animals provided that unnecessary sufferings for animals can be avoided) and animal rights (advocating the elimination of all forms of animal uses and giving animals equal ethical rights with humans) are increasingly conflated [41], and can lead to very serious issues for species conservation and SWM. The Italian gray squirrel provides a typical example. Although the gray squirrels introduced into Italy as pets pose a serious threat to the conservation of the native red squirrels, the plan to eradicate the gray squirrels ended in failure due to strong opposition from radical animal rights organizations and the public [42,43], and the uncertain publicity on social media exacerbated the failure of this management [44,45]. In Scotland, animal rights campaigners successfully focused the government to relocate rather than euthanize invasive hedgehogs that threatened rare bird colonies on Scottish islands against scientific advice and concerns about the welfare of hedgehogs following relocation [46]. Therefore, it is so significant to clarify the difference between animal welfare and animal rights because many students may not realize the consequence of advocating animal rights and possible harm to ecosystems and species conservation.

#### 4.2.2. Trophy Hunting

This research found that 72.1% of students believed that trophy hunting is cruel and inhumane (Figure 3). In China trophy hunting has been effectively banned since 2006 [47] due to public emotional opposition but a similar situation also exists in Western countries, where trophy hunting has been traditionally been more popular and esteemed until recently. For example, a recent study found over, the ethical views of Americans in the field of trophy hunting are changing, with the number of Americans opposed to trophy hunting increasing 7% from 2006 to 2016 [48].

Nearly half (49.3%) of students disagree that well-managed trophy hunting can be considered as a conservation management measure (Figure 3). Trophy hunting as a conservation management tool has been a controversial and polarized issue in recent years, and call for banning trophy hunting has become more intense with the widespread publicity of unethical and unregulated stories in the media around the world [49]. In fact, while unregulated trophy hunting and unscrupulous operators and hunters can threaten wildlife conservation efforts [50,51], trophy hunting regulated on a scientific management framework does promote wildlife conservation in local communities by checking the negative impacts on livelihoods of population growth and by providing funds for conservation [52,53]. Several studies have demonstrated that sustainably managed trophy hunting effectively reduced the illegal poaching of *Capra falconeri* in Tajikistan and Pakistan by providing economic incentives for local community residents, and helped stabilize *Capra falconeri* population [54,55]. Furthermore, a ban on trophy hunting will also not achieve the purpose of wildlife conservation, but even may cause the failure of wildlife conservation and management. For example, Kenya passed a trophy hunting ban in 1977, but the number of wildlife in Kenya did not recover as expected and declined by an average of 68% between 1977 and 2016. While the trophy hunting ban is not the only factors for the failure of wildlife conservation in Kenya, it is the fundamental cause [56].

Overall, our research indicated that most students lack an in-depth understanding of trophy hunting. Byrd demonstrated that people’s attitudes toward hunting depends on the purpose of hunting, with a higher degree of support for hunting for wildlife management than for trophies [57]. The narrative around trophy hunting may, therefore, need to be refined in future to better reflect the impacts of it on rural development and the role in the conservation and management of wildlife to help form a scientific and dialectical view of trophy hunting [58], rather than focusing only on the motivation of hunters.

### 4.3. Recommendations

#### 4.3.1. Environmental Education in Universities and Three-Year Colleges

Scientific knowledge and education are essential to promote wildlife conservation, and properly designed environmental education electives and course can bring about positive changes in people’s attitudes, knowledge and skills [59]. Although raising awareness of ethical and moral issues has enhanced student learning [60,61], some balance is required as and more effort is required in understanding how SWM and animal rights interact and are not always conflictual.

We suggest universities and three-year colleges should design the environmental education curriculum system scientifically to integrate the theory of SWM with ethics in order to refresh and adapted the appeal of SWM to the younger generation of Chinese students. Universities and three-year colleges can use a variety of platforms to carry out various short-term educational activities, such as special lectures, community activities, group discussions, etc., to train students to form positive, dialectical, independent thinking on issues regarding wildlife conservation management, especially “Animal Welfare and Rights” and “Trophy Hunting”. Targeted educational activities for groups with different demographics, should be designed so that the content of environmental education can be harmonized at different educational levels with a stronger focus on the theory of SWM [62].

#### 4.3.2. Popularization of the Theory of Sustainable Wildlife Management through Social Media

In addition to formal education, social media is another important way to influence attitudes. At present, social media, characterized by its simplicity and rapid dissemination, has provide Education Department (ED) an easy and convenient channel for the public to obtain conservation knowledge [63], and research has shown that social media has become the main source of public understanding of environmental information especially among young people [64]. However, social media may not always be able to convey the complexities of scientific findings nor the nuances of specific situation because of its inherent simplicity and rapidity, and this can lead to serious misunderstandings of conservation issues [44,64]. Therefore, we suggest that it is necessary for social media to popularize not only compassion and love, but also scientific knowledge and strategies in wildlife conservation narratives to ensure the sustainable development of wildlife conservation. Furthermore, professional scientists should actively disseminate scientific knowledge on social media platforms and interact with the public to help the spread of scientific conservation management concepts [65], such as the theory of SWM.

### 4.4. Limitations of the Study

Our study is the first systematic one to explore attitudes of students towards the SWM and wildlife conservation. Due to economic and time constraints, we only investigated the attitudes of students in nine universities and four three-year colleges in China in this study, which may not be representative of all Chinese students. However, through stratified sampling we attempted to improve the representativeness of the sample so as to ensure that the sample structure is consistent with the population. Furthermore, our study is seeking to establish the overall baseline about opinions and beliefs towards SWM. Due to the simple form of the questionnaire and insufficient of context, our results may not be transferred to specific contexts, but the focus of our research is to study the worldview of college students not asking people about specific context. More research is required in the future to provide a deeper analysis of opinions and beliefs towards SWM, for example through in-depth interviews, focus groups, and workshops.

## 5. Conclusions

Our study provides some insights into Chinese college students’ attitudes towards SWM and wildlife conservation. In general, students broadly support the SWM. Students have opposite perceptions of SWM on issues that can arouse strong emotions such as “Animal Welfare and Rights” and “Trophy Hunting”, which may have a close relationship with the ethical content of environmental education course and emotional publicity in social media. In addition, vegetarians, freshmen, and students who have taken environmental protection electives have less support for SWM. Therefore, further explorations on the role of environmental education system in universities and three-year colleges and social media in shaping attitudes of students towards sustainable wildlife management and conservation are needed in future studies, in order to maximize students’ understanding and support of SWM and promote sustainable development of wildlife conservation.

## Figures and Tables

**Figure 1 animals-10-01821-f001:**
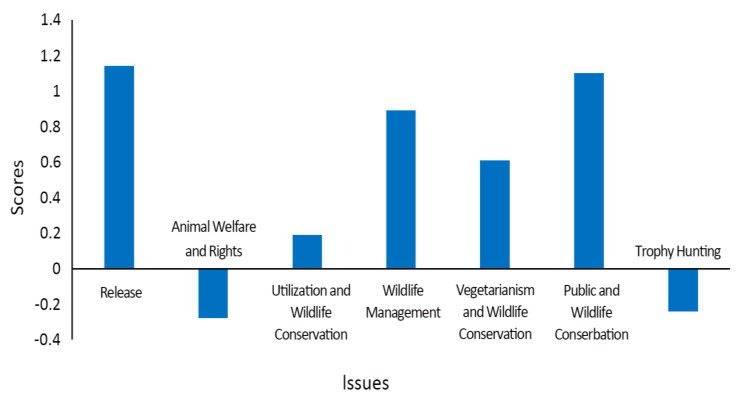
The mean scores of seven categories of issues on attitude questionnaires based on a Likert scale.

**Figure 2 animals-10-01821-f002:**
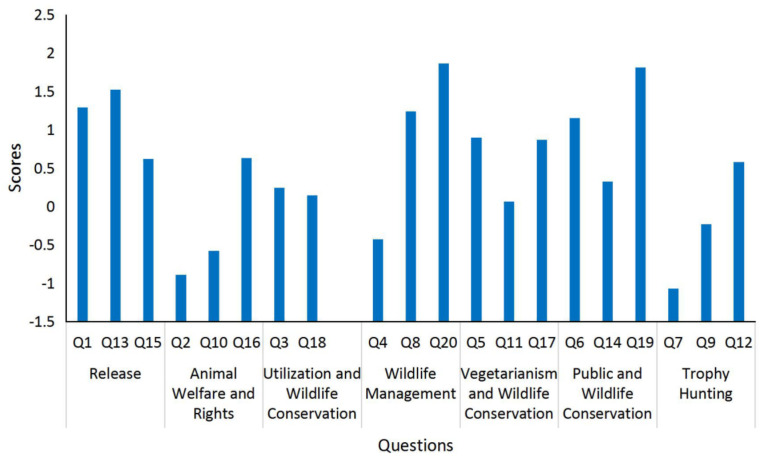
The mean scores of twenty questions on attitude questionnaires based on a Likert scale.

**Figure 3 animals-10-01821-f003:**
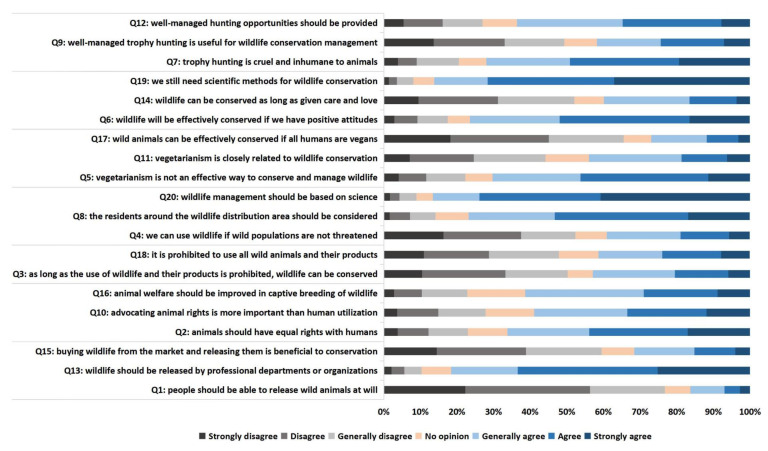
The statistical chart of option of respondents on twenty questions on attitude questionnaires.

**Figure 4 animals-10-01821-f004:**
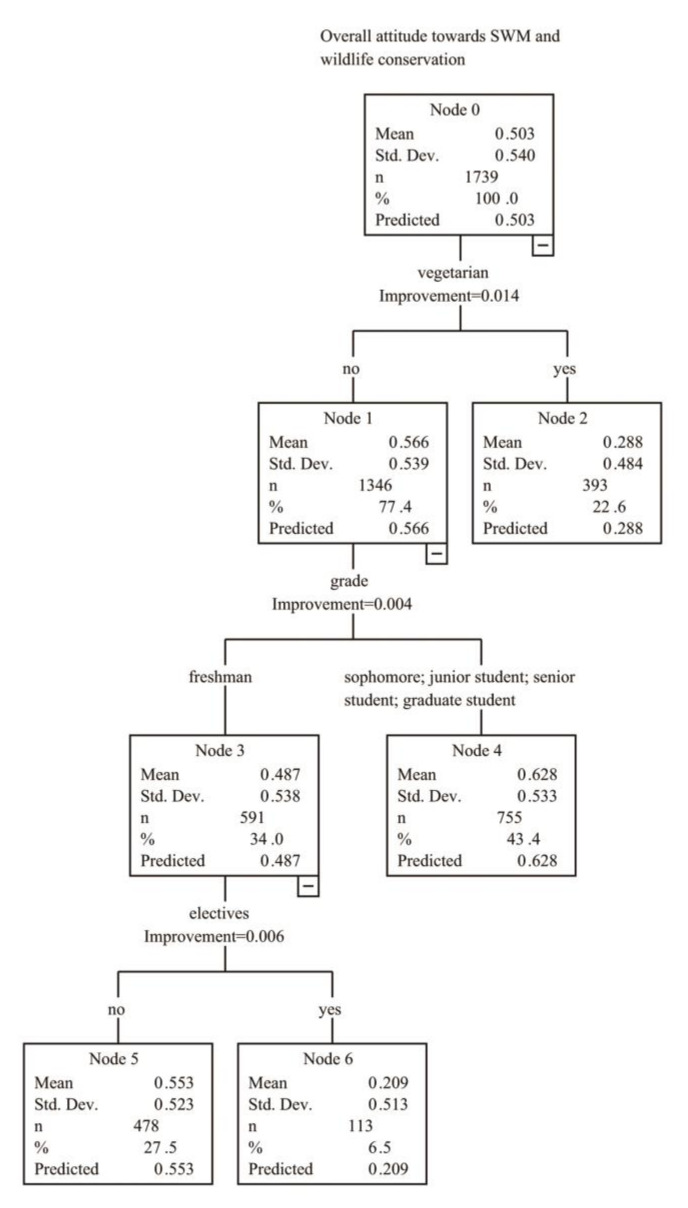
The classification and regression tree (CART) model for overall attitude toward SWM and wildlife conservation.

**Table 1 animals-10-01821-t001:** Samples of the universities and three-year colleges in China in the survey of attitudes toward sustainable wildlife management (SWM) and wildlife conservation.

Site	N	Class of University
East China Normal University	102	“Double First-Class” university
Nanchang University	99	
Northeast Forestry University	145	
North China University of Science and Technology	100	First-tier university
Hebei University of Science and Technology	200	
Changchun University of Traditional Chinese Medicine	100	
Lvliang University	202	Second-tier university
Taiyuan Normal University	199	
Jilin Agricultural Science and Technology University	150	
Shanxi Architectural College	94	Three-year college
Jiaozuo University	199	
Heze Vocational College	201	
Tianjin Coastal Polytechnic	200	

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
