# Peer review of "Current Societal Views about Sustainable Wildlife Management and Conservation: A Survey of College Students in China"

_animals, 2020, doi:10.3390/ani10101821_

Round 1
Reviewer 1 Report
- Line 72. It should be “common leopard”.
- Line 73. “Moschidae” should not be in italics.
- Lines 73-75. This part on Covid should be expanded and better explained, as it is important in influencing the relationship between humans and wildlife.
- Lines 97-98. Why college students are considered to be more concerned about animal welfare? May you provide some citations?
- Lines 98-106. I think that this part should be rephrased. Authors should clearly assess their predictions after their (well-explained) aims, so to put their paper in a more hypothesis-driven context and expand the interest of their work.
- Line 109. Is there an explanation for the selection of this study period?
- Methods are appropriate e and well-explained as well as the statistical part, but I think you should explain how did you treat privacy and sensitive information… for instance, was the questionnaire anonymous…ecc..
- Line 183. What is the Ethics Committee of Northeast Forestry University? May you explain for who is not familiar with the topic?
- I would like to suggest several papers to the authors, concerning human attitudes towards alien species, to be read and considered for their work: Archibald, J. L., Anderson, C. B., Dicenta, M., Roulier, C., Slutz, K., & Nielsen, E. A. (2020). The relevance of social imaginaries to understand and manage biological invasions in southern Patagonia. Biological Invasions, 1-17; Cerri J., Mori E., Zozzoli R., Gigliotti A., Chirco A., Bertolino S. (2020). Managing invasive Siberian chipmunks Eutamias sibiricus in Italy: a matter of attitudes and risk of dispersal. Biological Invasions 22: 603-616; Fitzgerald, G., Fitzgerald, N., & Davidson, C. (2007). Public attitudes towards invasive animals and their impacts. A summary and review of Australasian and selected international research. Invasive Animals Cooperative Research Centre (eds.): 1-57; La Morgia, V., Paoloni, D., & Genovesi, P. (2017). Eradicating the grey squirrel Sciurus carolinensis from urban areas: an innovative decision‐making approach based on lessons learnt in Italy. Pest Management Science, 73, 354-363; Sharp, R. L., Larson, L. R., & Green, G. T. (2011). Factors influencing public preferences for invasive alien species management. Biological Conservation, 144, 2097–2104.
- Results are quite clear, but figures are partly unreadable. I guess you have kept the main format of Microsoft Excel, but please use black labels, black axis and avoid that rectangles cover the written parts.
- Discussion is ok, but in this part you should clarify which of your predictions were fulfilled and which ones not.
Author Response
Comment 1: Line 72. It should be “common leopard”.
Reply 1: We have modified this in Ln 74. Thank you for the suggestion.
Comment 2: Line 73. “Moschidae” should not be in italics.
Reply 2: We have modified it in Ln 75.
Comment 3: Lines 73-75. This part on Covid should be expanded and better explained, as it is important in influencing the relationship between humans and wildlife.
Reply 3: We agree with the Reviewer 1 that Covid-19 is important to influence the relationship between humans and wildlife, so this part should be expanded and better explained. Now, we have changed this part in the text, and describe the content as follows (new text underlined):
Ln 75-81: Furthermore, the outbreak of the Covid-19 pandemic in 2020 has further affected the human-animal relationships in Chinese society. Although the origin of the Covid-19 has not yet been determined, it has caused a dramatic change of attitudes of public and government towards wildlife conservation and utilization. In order to promote wildlife conservation and prevent possible pandemics in the future, China enacted laws emphasizing the prohibition of consumption of all wildlife, and imposed more severe penalties for violations of the wildlife consumption ban.
Comment 4: Lines 97-98. Why college students are considered to be more concerned about animal welfare? May you provide some citations?
Reply 4: In this study, we believe that college students are more concerned about animal welfare than the older generation. This is because animal welfare is still an emerging phenomenon in China. Compared with the older generation, young college students are more open-minded and therefore be more quickly and inclusive when accepting new things (Su & Martens, 2017). Furthermore, since the concept of animal welfare was officially introduced in China in 1989, Chinese universities have actively opened animal welfare courses and carried out various activities (such as elective courses, lectures on animal welfare, broadcast films, sponsor debates etc.,) (Carpenter & Song, 2016), so the college students in China have better education opportunities and more knowledge related to animal welfare than other groups (Davey & Gareth, 2006; Su & Martens, 2017). Therefore, we are convinced that college students are more concerned about animal welfare than other groups, and they are also the most appropriate objects for this study. We have cited new references and detailed the reasons in the manuscript (new text underlined):
Ln 107-112: We focus on college students as they are more open-minded when accepting animal welfare, which is still an emerging phenomenon in China, and they have more educational opportunities related to animal welfare and therefore be considered to be more concerned about animal welfare than the older generation, and also because they will be the ‘influencers’ and leaders in China and their views about sustainable development and future wildlife management will be key to how SWM evolves in the future.
Comment 5: Lines 98-106. I think that this part should be rephrased. Authors should clearly assess their predictions after their (well-explained) aims, so to put their paper in a more hypothesis-driven context and expand the interest of their work.
Reply 5: We are very grateful for the Reviewer 1’s suggestions and agree that putting the aims before the hypothesis of the article will be clearer. We have now included a more detailed description of the aims and rephrased this part as follows (new text underlined):
Ln 104-116: Our key aim is to study the attitudes of college students to the theory of SWM and wildlife conservation in the context of rising support for animal welfare to identify specific areas of conflict between SWM and emerging attitudes of China’s college generation, and further investigate whether their demographics will affect their attitudes. We focus on college students as they are more open-minded when accepting animal welfare, which is still an emerging phenomenon in China, and they have more educational opportunities related to animal welfare and therefore be considered to be more concerned about animal welfare than the older generation, and also because they will be the ‘influencers’ and leaders in China and their views about sustainable development and future wildlife management will be key to how SWM evolves in the future. Specifically we explore the hypothesis that: college students as a whole are broadly supportive of the SWM (H1); demographic characteristics of college students (e.g vegetarian habits, grades, gender, major and interest in wildlife-related activities) can affect their degree of support for the SWM (H2).
Comment 6: Line 109. Is there an explanation for the selection of this study period?
Reply 6: Our research was completed with funding from the “2018 Key Project of Heilongjiang Province Educational Science Plan”, so the selection of the study period is based on this project.
Comment 7: Methods are appropriate and well-explained as well as the statistical part, but I think you should explain how did you treat privacy and sensitive information… for instance, was the questionnaire anonymous…ecc..
Reply 7: We are pleased the Reviewer 1 think that the Methods are appropriate. In fact, the survey was conducted anonymously, and before the survey, the interviewees and potential candidates were informed that the results of the survey were only used for academic research and all information provided would be kept strictly confidential. We detailed the changes to the text below (new text underlined):
Ln 192-194: All interviewees and potential candidates were anonymized and gave their oral informed consent for inclusion before they participated in the study, and the investigators assured them that all the information provided would be kept strictly confidential to ensure their privacy.
Comment 8: Line 183. What is the Ethics Committee of Northeast Forestry University? May you explain for who is not familiar with the topic?
Reply 8: The Northeast Forestry University Ethics Committee (NEFUEC) is an institution which examines whether the qualifications, experience, and technical capabilities of the researchers meet the experimental requirements, and whether the research plans are scientific and meet the requirements of ethical principles. Since our research mainly uses sociological methods to collect information about humans, the NEFUEC required us to provide a series of files such as project application and progress report which included specific methods to ensure the safety and privacy of interviewees, and any significant changes that may occur in the project.
Comment 9:I would like to suggest several papers to the authors, concerning human attitudes towards alien species, to be read and considered for their work:Archibald, J. L., Anderson, C. B., Dicenta, M., Roulier, C., Slutz, K., & Nielsen, E. A. (2020). The relevance of social imaginaries to understand and manage biological invasions in southern Patagonia. Biological Invasions, 1-17; Cerri J., Mori E., Zozzoli R., Gigliotti A., Chirco A., Bertolino S. (2020). Managing invasive Siberian chipmunks Eutamiassibiricus in Italy: a matter of attitudes and risk of dispersal. Biological Invasions 22: 603-616; Fitzgerald, G., Fitzgerald, N., & Davidson, C. (2007). Public attitudes towards invasive animals and their impacts. A summary and review of Australasian and selected international research. Invasive Animals Cooperative Research Centre (eds.): 1-57; La Morgia, V., Paoloni, D., &Genovesi, P. (2017). Eradicating the grey squirrel Sciurus carolinensis from urban areas: an innovative decision‐making approach based on lessons learnt in Italy. Pest Management Science, 73, 354-363; Sharp, R. L., Larson, L. R., & Green, G. T. (2011). Factors influencing public preferences for invasive alien species management. Biological Conservation, 144, 2097–2104.
Reply 9: Thanks for your suggestions. We have read all of the references and these references are very useful for us to understand the public's attitudes towards alien species and their influencing factors. We have cited Fitzgerald et al. (2007) in the Introduction (Ln 64), and cited Cerri et al. (2020), La Morgia et al. (2017), Sharp et al. (2011) in the Discussion (Ln 340-341).
Comment 10: Results are quite clear, but figures are partly unreadable. I guess you have kept the main format of Microsoft Excel, but please use black labels, black axis and avoid that rectangles cover the written parts.
Reply 10: We have redrawn the Figure 1, 2 and 3. Furthermore, we have modified the format of Figure 3 to better present the questions and the proportion columns horizontally.
Comment 11: Discussion is ok, but in this part you should clarify which of your predictions were fulfilled and which ones not.
Reply 11: We agree that whether the predications are fulfilled should be clarified in the Discussion, and we made the following changes as follows (new text underlined):
Ln 281-286: This research shows that Chinese college students are broadly supportive toward the theory of sustainable wildlife management (SWM), with an overall average score above zero (Mean = 0.503), which is consistent with our hypothesis (H1). However, the degree of support for SWM varies considerably depending on demographic characteristics, with vegetarianism, grade, and electives being the main factors that shape the attitudes of students. This supports our second hypothesis (H2).
References:
Carpenter, A. F. & Song, W. (2016) Changing Attitudes about the Weak: Social and Legal Conditions for Animal Protection in China. CRITICAL ASIAN STUDIES, 48, 380-399.
Davey & Gareth (2006) Chinese University Students' Attitudes Toward the Ethical Treatment and Welfare of Animals. Journal of Applied Animal Welfare ence, 9, 289-297.
Su, B. & Martens, P. (2017) Public attitudes toward animals and the influential factors in contemporary China. ANIMAL WELFARE, 26, 239-247.

Reviewer 2 Report
The objective of the manuscript is to assess changes in views about sustainable wildlife management. However, the study assessed views exclusively of students, so how can the study assess changes if there is no comparison with other groups (e.g. e.g. university alumni or wildlife management professionals)? I therefore doubt that this study can make any conclusions about changes in views and this should also be clear beginning at the title. I also have concerns about the interpretation. To some of the 20 questions of the questionnaire, there are clear standpoints for wildlife managers (e.g. “people can freely release wild animals”), but to others (e.g. question 2 “animals should have equal rights with humans”) there is no generally right or wrong answer, this is an ethical question. Some of the questions would in some occasions be beneficial for wildlife management, in others not (e.g. question 9. Trophy hunting can be sustainable or not, it can help conservation in some cases, in others is not beneficial). If you think there is only one right answer, then you should explain in detail which answer would fully be in line with sustainable wildlife management. In my opinion, the objective of sustainable wildlife management is to assure that the use of wildlife (if wanted) is sustainable, but it is no goal of wildlife management to use as much wildlife as possible in general. So how can you actually give scores? As presented now, you give the impression that a higher score means that people better understand wildlife management, but I do not think this is correct. Besides, it is not appropriate to calculate averages for ordinal variables such as your response variable. Therefore Figure 1 is not an appropriate form of presenting the results, Figure 3 is much more useful, but you should include all 7 categories. It is also difficult to assess the results without the questions, so I suggest to write the question here horizontally in full and present the proportion columns horizontally. I therefore think you need to edit the presentation of results. If you want to “score”, you also need to indicate (with justification) which answer would be in line with the viewpoint of a wildlife manager. Otherwise, I think you need to write in a more objective way and just present the results without judging them.
Author Response
Reply:
- Reply 1: How can the study assess changes if there is no comparison with other groups?
We thank Reviewer 2 for the comments, and apologize for not clarifying this point in the original manuscript. In fact, Chinese society’s attitudes towards the sustainable wildlife management has been changing in the past few years, and this is mainly reflected in the comprehensive understanding of the relationship between wildlife conservation and sustainable utilization which is an important aspect of sustainable management (Morgera & Wingard, 2009). Specifically, Chinese society has always adhered to the "reformed anthropocentrism" conservation management concept, which acknowledges the intrinsic value of wildlife, but also emphasizes that the conservation and sustainable use should be effectively combined (Zinn & Shen, 2007). However, in recent years, with the increasing public concern about animal welfare, animal treatment and even animal rights, Chinese society’s concern and love for individual animals has increased rapidly (Lu et al., 2013), and more and more people have begun to advocate wildlife management strategies that prohibit any use (ZHOU et al., 2016). In this study, we did not quantitatively observe the changes in the view of sustainable wildlife management, but qualitatively believe that the changes have occurred. Therefore, based on the very useful comments of the Reviewer 2, we changed the title of the manuscript to better fit the subject of this research (new text underlined).
Ln 2-4: Current Societal Views about Sustainable Wildlife Management and Conservation: A Survey of College Students in China
- Reply 2: How can you give scores on issues such as Animal Welfare and Rights and Trophy Hunting?
Sustainable wildlife management (SWM) is the sound management which aims to conserve species populations, while meeting human socio-economic needs through sustainable use (Van Vliet et al., 2015). The scientific combination of wildlife conservation and sustainable utilization is the requirement and basic concept of SWM (Webb, 2002; Morgera & Wingard, 2009), and an objective and rational understanding of the relationship between wildlife conservation and sustainable utilization is therefore an important basis for understanding of the SWM.
In fact, “Animal Welfare and Rights” is not only an ethical issue, but also an important aspect of understanding the relationship between wildlife conservation and sustainable utilization. Animal welfare and animal rights are very different in understanding the relationship between conservation and sustainable utilization. Specifically, animal welfare advocates believe that humans should apply ethical principles to animals, and can reasonably use or scientifically manage animals provided that the suffering of affected animals can be justified, measured and minimized (David, 2016; HAMPTON & TEH-WHITE, 2019). Animal rights, on the other hand, are opposed to killing or harming any individual animal for any reason based on the philosophical reason that humans and animals should be equal (Regan & Singer, 1989). However, as animal rights organizations use social media to promote a lot of animal lovely stories, animal welfare and animal rights are increasingly conflated (Allen, 2018), and can lead to very serious issues for sustainable wildlife management and conservation. For example, in Australia, animal rights organizations use the protection of animal welfare as an excuse to lobby the public not to manage populations that are in fact over-abundant and harmful species that threaten Australia’s biodiversity and economy (Banks, 2005). Therefore, animal welfare and animal rights may seem to be a moral issue, but in fact they are a deep issue affecting the sustainable wildlife management.
In this study, question 2“animals should have equal rights with humans” and question 10 “Advocating animal rights is more important than utilization of wildlife by men” reflect a view of animal rights. The lower the score, the more the respondents support animal rights. Many scholars and organizations have recognized that the view of animal rights runs counter to the sustainable wildlife management (Banks, 2005; Trull & L., 2015; HAMPTON & TEH-WHITE, 2019). In terms of the question 16 "Animal welfare should be improved in captive breeding of wildlife”, the higher the score, the more the respondents support this view. Improving the welfare of captive breeding of wildlife meets the requirements of sustainable wildlife management. This is because captive breeding is an important conservation intervention for the sustainable wildlife management (Huber et al., 2019), and positive indicators of animal welfare are essential components of successful captive breeding programs (Greggor et al., 2018). Ensuring optimum animal welfare is important for maintaining healthy and sustainable populations because they are correlated with reductions of physiological indicators of stress, occurrence of health problems, and increases in the breeding efficiency (Greggor et al., 2018; Khattak et al., 2019).
Regarding the issue of Trophy Hunting, we agree with your opinion that trophy hunting is beneficial to wildlife management in some cases, but harmful in other cases. In fact, although this question was not reflected in the questionnaire, during the survey, we have explained to the interviewees and potential candidates that the trophy hunting mentioned in the questionnaire refers to hunting activities based on a standardized management framework.
Overall, based on the useful comments of the Reviewer 2, we have also reflected on the methodology and pointed out the methods for future improvement in the Discussion of the manuscript (new text underlined).
Ln 414-419: Furthermore, our study is seeking to establish the overall baseline about attitudes and beliefs towards SWM. However, the simple form of the questionnaire and the insufficient of context have made us lose the opportunity to study the interviewees’ thoughts in depth. Therefore, more research is required in the future to provide a deeper analysis of support for SWM in the context of rising support for animal rights and welfare, for example through in-depth interviews, focus groups and workshops.
- Reply 3: Edit the presentation of results and Figures.
According to the analysis of Reply 2, We believe that presenting the results in the form of scoring is an appropriate choice. Therefore Figure 1 and Figure 2 is an appropriate form of presenting the results and can be a good indicator of respondents' attitudes towards sustainable wildlife management. For Figure 3, we agree that it is difficult to assess the results without the questions, so we redrew Figure 3 to show the proportion columns and questions horizontally to better show the results.
References:
Allen, D. (2018) Book Review: The Gospel of Kindness. Animal Welfare and the Making of Modern America, by Janet M. Davis. Agricultural History Review, 65, 348-350.
Banks, P. B. (2005) Animal-rights zealots put wildlife welfare at risk. NATURE,.
David, M. (2016) Moving beyond the “Five Freedoms” by Updating the “Five Provisions” and Introducing Aligned “Animal Welfare Aims”. Animals, 6, 59.
Greggor, A. L., Vicino, G. A., Swaisgood, R. R., Fidgett, A., Brenner, D., Kinney, M. E., Farabaugh, S., Masuda, B. & Lamberski, N. (2018) Animal Welfare in Conservation Breeding: Applications and Challenges. Frontiers in Veterinary Science, 5.
HAMPTON, J. O. & TEH-WHITE, K. (2019) Animal Welfare, Social License, and Wildlife Use Industries. The Journal of Wildlife Management, 83, 12-21.
Huber, N., Marasco, V., Painer, J., Vetter, S. G. & Walzer, C. (2019) Leukocyte Coping Capacity: An Integrative Parameter for Wildlife Welfare Within Conservation Interventions.
Khattak, R. H., Liu, Z. & Teng, L. (2019) Development and Implementation of Baseline Welfare Assessment Protocol for Captive Breeding of Wild Ungulate—Punjab Urial (Ovis vignei
punjabiensis, Lydekker 1913). Animals, 9, 1102.
Lu, J., Bayne, K. & Wang, J. (2013) Current Status of Animal Welfare and Animal Rights in China. Alternatives to Laboratory Animals, 41, 351-357.
Morgera, E. & Wingard, J. (2009) Principles for developing sustainable wildlife management laws. Cic Technical,.
Regan, T. & Singer, P. (1989) Animal rights and human obligations.
Trull & L., F. (2015) PETA undermines science and scientists. SCIENCE, 347, 834.
Van Vliet, N., Gomez, J., Quiceno-Mesa, M. P., Escobar, J. F., Andrade, G., Vanegas, L. A. & Nasi, R. (2015) Sustainable wildlife management and legal commercial use of bushmeat in Colombia: the resource remains at the cross-road. INTERNATIONAL FORESTRY REVIEW, 17, 438-447.
Webb, G. J. W. (2002) Conservation and sustainable use of wildlife - An evolving concept. Pacific Conservation Biology, 8, 12-26.
ZHOU, X., WAN, X., JIN, Y. & ZHANG, W. (2016) Concept of scientific wildlife conservation and its dissemination. Zoological Research,.
Zinn, H. C. & Shen, X. S. (2007) Wildlife Value Orientations in China. 12, 331-338.

Round 2
Reviewer 2 Report
The paper has somewhat improved in the revision, my major concern is however that I do not think that the questions are unambiguous. In my opinion, wildlife managers would not necessarily answer all questions with “strongly agree” or “strongly disagree”, because often the answer should rather be “it depends”. For example, there is no “correct” answer to question 5, “Vegetarism is not an effective way to conserve wildlife”, because it can contribute to decrease hunting pressure on some species. As there are many such ambiguous questions (see detailed comments), I do not think that the “score” is appropriate as it does not depend on knowledge. You should rather provide the specific answers (strongly agree, agree…) in Figure 3. Given that the questions are not fully representative for a wildlife manager’s opinion, you should also discuss the possibility that the questions might not exactly provide the opinion of the interviewed person. For example, a wildlife manager might answer the vegetarism question with “strongly agree” if understood as a general measure that would always apply, but might also answer with “agree” or “generally agree” to account for the contribution vegetarism can make to conservation. I therefore think you should not score the opinions as there is no definite answer.
Detailed comments:
Line 124. Not clear what you mean by a “response efficiency of 88%”, please define.
Table 1. It is not completely clear which university belongs to which class.
Line 130. “had concerned about…” is not clear, do you mean “had concerns about wildlife conservation”? Which concerns? Please reword the sentence to make this clear.
Line 139. “Q1: People can freely release wild animals”. The question is not clear. Do you mean here “people should freely release wild animals”? Or is this a legal knowledge question. “Are you allowed to freely release wild animals”? Please make clear which sense had your question.
Line 140 “departments” is probably better here.
Line 142. In my opinion, these questions are independent of wildlife management. A wildlife manager can either agree or not degree with this. Animal welfare also does not necessarily render captive breeding less efficient.
Line 145-146. Question Q3: prohibition of use of a species can be a management tool and be efficient in conservation, so a wildlife manager can here agree in specific cases and disagree in others. There is no general SWM opinion about this question.
Line 146-147. Q18 is unclear, is this an opinion question or a legal knowledge question? I guess you mean here “wild animals and their products should not be used…”
Line 149. Probably “standard of living” would be better here.
Line 150-151. Q20 is ambiguous. What does it mean to “scientifically manage”? It could be anything. This needs a better definition. E.g. “wildlife management should be based on science”.
Line 152. Q5 is ambiguous. Vegetarianism can of course be a tool for conservation and wildlife management, if you eat less, there is less also less hunting pressure. So how can this question contribute to SWM? In your opinion, should wildlife management always aim for maximal exploitation? I do not think this is the goal of wildlife management. Q17 is also not unambiguous as probably some species could be preserved exclusively through stop of human consumption.
Line 155. Q6 is also ambiguous because a positive attitude can certainly contribute to conservation, so people might not completely disagree here. In my opinion such a question does not allow the interviewed people to give a precise opinion.
Line 159. Q7 is also ambiguous. It is not that all wildlife managers would or should support trophy hunting. Not clear what “standardized trophy hunting” is? Please define. This question is also ambiguous as for some species trophy hunting can be a conservation measure (e.g. introduced invasive species), but for other not.
Line 194. This whole subchapter is not a result because it only describes the composition of the people interviewed, it should go to the methods
Figure 1 and figure 2. I do not see the value of this figure as a wildlife manager should not always fully agree or disagree with the ambiguous questions.
Line 207. Why do you report here only the answers of college students and not of the universities? Or do you mean here all students? In this case you need to change the definition of college in Table 1.
Figure 3. this figure is now much more informative, but it would be good to show rather all 7 categories instead of reducing it to 3. You could do this e.g. by gradually changing the brightness of blues and greys. “not bothered” does not seem to be a good term, maybe “undecided” or “no opinion”?
Author Response
Comments and Suggestions for Authors
The paper has somewhat improved in the revision, my major concern is however that I do not think that the questions are unambiguous. In my opinion, wildlife managers would not necessarily answer all questions with “strongly agree” or “strongly disagree”, because often the answer should rather be “it depends”. For example, there is no “correct” answer to question 5, “Vegetarism is not an effective way to conserve wildlife”, because it can contribute to decrease hunting pressure on some species. As there are many such ambiguous questions (see detailed comments), I do not think that the “score” is appropriate as it does not depend on knowledge. You should rather provide the specific answers (strongly agree, agree…) in Figure 3. Given that the questions are not fully representative for a wildlife manager’s opinion, you should also discuss the possibility that the questions might not exactly provide the opinion of the interviewed person. For example, a wildlife manager might answer the vegetarism question with “strongly agree” if understood as a general measure that would always apply, but might also answer with “agree” or “generally agree” to account for the contribution vegetarism can make to conservation. I therefore think you should not score the opinions as there is no definite answer.
Reply:
We thank the reviewer for your comments and appreciate the careful assessment of the manuscript. For the ambiguous questions that you pointed out, some questions are due to inaccurate translation, which leads to the failure to express its true meaning in English, and we apologize for this. For other questions, in fact, although these questions are not placed in a specific context, this does not affect our research. This is because we are trying to gauge overall opinion/beliefs about issue related to SWM. We appreciate the importance of context but we are not asking people about specific context but rather about their ‘worldview’ because world view is becoming more and more important to understanding conservation/SWM conflicts. Therefore, in order to make readers aware of this, we have added some content to the Discussion (Ln 417-422) and also explained why these questions are suitable for this research in the “Detailed comments”. Furthermore, there are many studies that have adopted similar methods to our approach. The following are some article examples of using similar methods.
- Liordos, V., Kontsiotis, V. J., Anastasiadou, M., & Karavasias, E. (2017). Effects of attitudes and demography on public support for endangered species conservation. Science of the Total Environment, 595, 25-34.
- Cerri, J., Mori, E., Vivarelli, M., & Zaccaroni, M. (2017). Are wildlife value orientations useful tools to explain tolerance and illegal killing of wildlife by farmers in response to crop damage? European Journal of Wildlife Research, 63(4), 70.
- Manfredo, M. J., Teel, T. L., & Dietsch, A. M. (2016). Implications of human value shift and persistence for biodiversity conservation. Conservation Biology, 30(2), 287-296.
- Chase, L. D., Teel, T. L., Thornton-Chase, M. R., & Manfredo, M. J. (2016). A comparison of quantitative and qualitative methods to measure wildlife value orientations among diverse audiences: A case study of Latinos in the American Southwest. Society & natural resources, 29(5), 572-587.
- Zainal Abidin, Z. A., & Jacobs, M. H. (2016). The applicability of wildlife value orientations scales to a Muslim student sample in Malaysia. Human Dimensions of Wildlife, 21(6), 555-566.
- Hermann, N., Voß, C., & Menzel, S. (2013). Wildlife value orientations as predicting factors in support of reintroducing bison and of wolves migrating to Germany. Journal for Nature Conservation, 21(3), 125-132.
- Butler, J. S., Shanahan, J. E., & Decker, D. J. (2001). Wildlife attitudes and values: A trend analysis.
- Manfredo, M., Teel, T., & Bright, A. (2003). Why are public values toward wildlife changing? Human Dimensions of wildlife, 8(4), 287-306.
Detailed comments:
Comment 1: Line 124. Not clear what you mean by a “response efficiency of 88%”, please define.
Reply 1: The “response efficiency” is completion rate. Completion rate is a measure used to describe data collection procedure and quality of data collected. It is number of respondents who fully completed the survey divided by the number of respondents who entered the survey (Encyclopedia, 2016; Bose, 2020). In our study, 1991 persons were asked to take part, 1977 of them accepted and 1739 actually completed questionnaires then the completion rate is 88% (CR=1739/1977). We have changed this term in the text (new text underlined):
Ln 127-128: After descriptive statistical preliminary analysis, incomplete questionnaires were eliminated there was 1739 valid questionnaires with a completion rate of 88%.
Comment 2: Table 1. It is not completely clear which university belongs to which class.
Reply 2: Based on your useful comment, we redrew the Table 1 to better display the class of universities and colleges (Ln 129).
Comment 3: Line 130. “had concerned about…” is not clear, do you mean “had concerns about wildlife conservation”? Which concerns? Please reword the sentence to make this clear.
Reply 3: We apologize for the inaccurate wording in the manuscript. In fact, we want to ask the respondents whether they have paid attention to information related to wildlife conservation, for example, have they read news or publicity about wildlife conservation. We have reworded the sentence as below (new text underlined):
Ln 132-138: In the first part, we ascertained the demographic characteristics of the respondents, including gender, grade, major (agriculture, science, engineering, medicine, economics, management, law, literature, fine arts), and recorded whether the respondent was a vegetarian (abbr. vegetarian), whether the respondent had ever paid attention to information related to wildlife conservation (abbr. attention), whether the respondent taken environmental protection electives during university/college (abbr. electives) or participated in wildlife related activities (abbr. activity).
Comment 4: Line 139. “Q1: People can freely release wild animals”. The question is not clear. Do you mean here “people should freely release wild animals”? Or is this a legal knowledge question. “Are you allowed to freely release wild animals”? Please make clear which sense had your question.
Reply 4: We apologize for the ambiguity caused by inaccurate translation in Q1. In this research, Q1 is an opinion question about whether people should be able to release wild animals at will. We believe that people cannot release wild animals at will, but in order to avoid the model effect of the questionnaire and enable respondents to think fully, we deliberately set the Q1 from reverse angle to better explore the respondent’s attitude towards the wildlife release (strongly disagree= +3, disagree= +2, generally disagree=+1, indifferent= 0, generally agree = -1, agree = -2, strongly agree = -3. The lower the score, the more the respondents support random release). We have retranslated Q1 to make it more clearly (new text underlined):
Ln 145: People should be able to release wild animals at will.
Comment 5: Line 140 “departments” is probably better here.
Reply 5: We have modified this and thanks for your suggestion (Ln 146).
Comment 6: Line 142. In my opinion, these questions are independent of wildlife management. A wildlife manager can either agree or not degree with this. Animal welfare also does not necessarily render captive breeding less efficient.
Reply 6: Thanks for your careful comments. In fact, in this study, we are not studying the opinions of wildlife managers, but the attitudes of college students, and the relationship between animal welfare and animal rights and wildlife management has been widely recognized by many scientists.
For example, Hampton (2019) indicated that animal rights erode public support for otherwise sustainable wildlife industries and pose a great threat to wildlife management (HAMPTON & TEH-WHITE, 2019). Trull (2015) pointed out that animal rights organizations are strongly opposed to scientific research on laboratory animals, severely weakening the inevitable way to obtain scientific information required for wildlife management, and eroding the ability of management of endangered species (Mcmahon et al., 2012; Trull & L., 2015). Genovesi (2001) confirmed that due to the strong opposition of animal rights organizations, the Italian government lost the best time to eradicate the invasive species, leading to the failure of management plan of invasive species (Genovesi & Bertolino, 2001). Carson (2012) indicated that radical animal rights organizations have caused thousands of crimes in the United States, which have had a serious impact on the wildlife management and even the social and economic order in the United States (Carson et al., 2012). Furthermore, Paquet (2010) believed that animal welfare is the ethical foundation of wildlife conservation and management and needs to be taken seriously (Paquet & Darimont, 2010). Mellor (2016) indicated that animal welfare is closely related to wildlife management, and wildlife management departments should make extensive use of available information to improve animal welfare as much as possible (David, 2016). Dubois (2013) pointed out that the application of animal welfare science in the field of conservation and management can improve the humaneness of practices and generate support for wildlife management plans (Dubois & Harshaw, 2013).
Therefore, animal welfare and animal rights do not seem to be directly related to wildlife management, but the indirect relationship that exists is an important aspect that must be considered in wildlife management.
Comment 7: Line 145-146. Question Q3: prohibition of use of a species can be a management tool and be efficient in conservation, so a wildlife manager can here agree in specific cases and disagree in others. There is no general SWM opinion about this question.
Reply 7: Thanks for your comment and careful review of the manuscript. Regarding Q3, what we expressed in the questionnaire is "As long as the use of wild animals and their products is prohibited, wild animals can be effectively conserved", but due to the inaccurate translation from Chinese to English, the absolute view hidden in the question is not well expressed. This absolute view is contrary to the SWM opinion. This is because in the field of wildlife management, habitat destruction, climate change, human over-utilization, and invasion of alien species are considered to be main factors for the decline of wildlife populations (Oommen et al., 2019), and a comprehensive wildlife management strategy is the requirement for SWM (Morgera & Wingard, 2009). In some specific cases, the prohibition of wildlife use can indeed be used as a management tool, but only relying on prohibition of use not only fails to achieve the purpose of effective conservation (Ogutu et al., 2016), it is also contrary to the requirement of SWM. Therefore, based on your useful comment, we have retranslated Q3 to make it more accurately (new text underlined):
Ln 151-152: As long as the use of wild animals and their products is prohibited, wild animals can be effectively conserved.
Comment 8: Line 146-147. Q18 is unclear, is this an opinion question or a legal knowledge question? I guess you mean here “wild animals and their products should not be used…”
Reply 8: Thanks for your suggestion. Q18 is an opinion question, and we reworded the Q18 to make it more clearly (new text underlined):
Ln 152-153: All wild animals and their products, including captive breeding of wild animals and their products, should not be used.
Comment 9: Line 149. Probably “standard of living” would be better here.
Reply 9: Thanks for your suggestion and we have modified this (Ln 155).
Comment 10: Line 150-151. Q20 is ambiguous. What does it mean to “scientifically manage”? It could be anything. This needs a better definition. E.g. “wildlife management should be based on science”.
Reply 10: We apologize for the unclear expression in the Q20 caused by the inaccurate translation. In fact, “scientifically manage” here means that wildlife management requires scientific methods and a comprehensive understanding of biological and ecological factors, such as species’ habitats, population sizes, migration routes, and population demographics. Based on your useful comment, we have retranslated Q20 to make it more accurately (new text underlined):
Ln156-157: Wildlife management should be based on science.
Comment 11: Line 152. Q5 is ambiguous. Vegetarianism can of course be a tool for conservation and wildlife management, if you eat less, there is less also less hunting pressure. So how can this question contribute to SWM? In your opinion, should wildlife management always aim for maximal exploitation? I do not think this is the goal of wildlife management. Q17 is also not unambiguous as probably some species could be preserved exclusively through stop of human consumption.
Reply 11: Thank the reviewer for your comments. Regarding Q5, vegetarianism can contribute to wildlife conservation in some certain situations, but it cannot be used as an effective tool for conservation and wildlife management. This is because vegetarianism can only be treated as personal choice, not a management standard (Knezevic, 2009). The choice of vegetarian by humans can indeed reduce the hunting pressure, promoting the growth of population of species and having some effect on conservation, but it is not necessarily beneficial to wildlife management which aims to keep the population at a healthy and appropriate level on the basis of scientific, technical and traditional knowledge. For example, due to reductions in hunting and natural predators (C Té et al., 2004), the population of white-tailed deer in North America has exploded to 2-4 times than its historical level (BIONDI et al., 2011), and the overabundance of white-tailed deer has negatively impacted forest health, ecosystem balance, human activity and the health of local deer populations (BIONDI et al., 2011). In fact, in this situation, hunters can actually partially replace natural carnivores, and thus play a role in managing and conserving white-tailed deer through proper hunting.
Regarding Q17, the "we" in the questionnaire actually refers to "all the human beings", and the true meaning of this question is " Wild animals can be effectively conserved if all the human beings are vegans", but due to translation errors, this precondition is not well expressed in English. At the same time, it is worth noting that in this question, we are referring to vegans who refuse to eat all animal meat and animal products, not semi-vegetarians who just reduce their meat intake. In fact, wild animals cannot be effectively conserved if all the human beings become vegans. This is because, first of all, meat is a strong driving force for the human evolution (Pereira & Vicente, 2013; Williams & Hill, 2017). Vegetarian food generally provides much less energy and nutrition than meat (Mann, 2000; Mann, 2013), and the human brain would not have evolved so well without meat eating (Mann, 2000; Williams & Hill, 2017). Therefore, in order to maintain human civilization, it is unscientific and unrealistic for all humans to be vegans. Secondly, humans, like many animals, are a member of the food chain, the predation relationship with animals meeting the requirements of ecological balance. Blindly cutting off this relationship will destroy the balance of the ecosystem (Fox, 1986).
Therefore, we have retranslated Q17 to make it more clearly. Furthermore, in Q5, our "conserve" refers to "conserve and manage", but due to inaccurate translation, it has not been well expressed, so we also retranslated Q5 (new text underlined):
Ln 158-160: Q5: Vegetarianism is not an effective way to conserve and manage wildlife; Q17: Wild animals can be effectively conserved if all the human beings are vegans;
Comment 12: Line 155. Q6 is also ambiguous because a positive attitude can certainly contribute to conservation, so people might not completely disagree here. In my opinion such a question does not allow the interviewed people to give a precise opinion.
Reply 12: Thanks for the reviewer’s careful review. In fact, in this study, we agree that a positive attitude will contribute to the wildlife conservation and sustainable wildlife management. Therefore, we set Q6 from a forward angle and assigned positive value to Q6 on the Likert seven-point scale (strongly disagree= -3, disagree= -2, generally disagree= -1, indifferent= 0, generally agree = +1, agree = +2, strongly agree = +3. The higher the respondent’s score on Q6, the more the respondent supports this opinion). People can choose the answer according to their own opinions.
Comment 13: Line 159. Q7 is also ambiguous. It is not that all wildlife managers would or should support trophy hunting. Not clear what “standardized trophy hunting” is? Please define. This question is also ambiguous as for some species trophy hunting can be a conservation measure (e.g. introduced invasive species), but for other not.
Reply 13: Trophy hunting is a form of managed hunting whereby specific animals are targeted for the primary purpose of harvesting body part(s) such as skin, antlers and head (Lindsey et al., 2016; Cooney et al., 2017). “Standardized trophy hunting” refers to well-managed and sustainable trophy hunting, which aims to increase the value of wildlife and the habitats it depends on, generating critically needed incentives and revenue for government, private and community landowners to maintain and restore wildlife as a land use and to carry out conservation actions (including anti-poaching interventions) (Muposhi et al., 2016; Cooney et al., 2017).
Trophy hunting takes place in a wide range of governance, management and ecological contexts and, accordingly, its impacts on conservation vary enormously. In some cases, due to weak governance, corruption, lack of transparency, excessive quotas, illegal hunting, etc., trophy hunting may be detrimental to wildlife conservation (Crosmary et al., 2015; Lindsey et al., 2016). But under well-managed conditions trophy hunting can be used as an important conservation intervention. For example, before establishing a hunting plan, the best science and technology should be used to assess the resources of the hunting population (examples might include counts or indices of population performance such as sighting frequencies, spoor counts) and establish a hunting indices (examples might include trophy size, animal age, hunting success rates and catch per hunting effort) (SSC, 2012); involves adaptive management of hunting quotas and plans in line with results of resource assessments and/or monitoring of indices (SSC, 2012; Wanger et al., 2017); trophy hunting should be based on laws, regulations and quotas that are transparent, clear and are periodically reviewed and updated (SSC, 2012); a monitoring system should be designed to effectively monitor population trends and status (CITES).
In our research, well-managed trophy hunting refers to hunting activities based on the above well-managed conditions. Therefore, for Q9, the trophy hunting we mentioned does not involve the situation that "trophy hunting can be a conservation measure for some species, but for others not", because species that are not suitable for trophy hunting have already been excluded due to the preconditions. Regarding Q7, well-managed trophy hunting needs to be incorporated into the consideration of sustainable wildlife management, which has been recognized by many scientists and international conservation organizations (Minin et al., 2016; Muposhi et al., 2016; Cooney et al., 2017; Wanger et al., 2017), and it is also worth noting that we are studying the attitudes of college students towards SWM, not wildlife managers, so we believe this is not an ambiguous issue.
Comment 14: Line 194. This whole subchapter is not a result because it only describes the composition of the people interviewed, it should go to the methods
Reply 14: Thanks for your suggestion. But we did not transfer this subchapter to the methods as suggested, because most articles published in Animals include this part in the results. The following are some article examples of describing the composition of respondents in the results.
- Sinclair, M., Yan, W., & Phillips, C. J. (2019). Attitudes of Pig and Poultry Industry Stakeholders in Guandong Province, China, to Animal Welfare and Farming Systems. Animals, 9(11), 860.
- Sharma, A., Schuetze, C., & Phillips, C. J. (2019). Public attitudes towards cow welfare and cow shelters (gaushalas) in India. Animals, 9(11), 972.
- Sharma, A., Schuetze, C., & Phillips, C. J. (2020). The Management of Cow Shelters (Gaushalas) in India, Including the Attitudes of Shelter Managers to Cow Welfare. Animals, 10(2), 211.
- Fröhlich, N., Sells, P. D., Sommerville, R., Bolwell, C. F., Cantley, C., Martin, J. E., ... & Coombs, T. (2020). Welfare Assessment and Husbandry Practices of Working Horses in Fiji. Animals, 10(3), 392.
Comment 15: Figure 1 and figure 2. I do not see the value of this figure as a wildlife manager should not always fully agree or disagree with the ambiguous questions.
Reply 15: Based on the above analysis, we believe that the questions can represent the overall opinions of college students about issues related to the sustainable wildlife management. Therefore, we believe that Figure 1 and Figure 2 are appropriate forms of displaying results.
Comment 16: Line 207. Why do you report here only the answers of college students and not of the universities? Or do you mean here all students? In this case you need to change the definition of college in Table 1.
Reply 16: We apologize for the inaccurate wording in the Table 1, and the college students in the manuscript refers to all students. Based on your useful comment, we changed the “College” in Table 1 to “Three-year college” to make it more clearly.
Comment 17: Figure 3. this figure is now much more informative, but it would be good to show rather all 7 categories instead of reducing it to 3. You could do this e.g. by gradually changing the brightness of blues and greys. “not bothered” does not seem to be a good term, maybe “undecided” or “no opinion”?
Reply 17: Thanks for your suggestion, and we have redrawn the Figure 3 based on your useful comment.
References:
BIONDI, K. M., BELANT, J. L., MARTIN, J. A., DEVAULT, T. L. & WANG, G. (2011) White-Tailed Deer Incidents With U.S. Civil Aircraft. WILDLIFE SOCIETY BULLETIN, 35, 303-309.
Bose, S. (2020) Survey completion rate vs. response rate: Differences explained. Zoho Survey,.
C Té, S. D., Rooney, T. P., Tremblay, J. P., Dussault, C. & Waller, D. M. (2004) Ecological Impacts of Deer Overabundance. Annual Review of Ecology Evolution & Systematics, 35, 113-147.
Carson, J. V., LaFree, G. & Dugan, L. (2012) Terrorist and Non-Terrorist Criminal Attacks by Radical Environmental and Animal Rights Groups in the United States, 1970–2007. TERRORISM AND POLITICAL VIOLENCE, 24, 295-319.
CITES Trade in hunting trophies of species listed in Appendix I or II.
Cooney, R., Freese, C., Dublin, H., Roe, D., Mallon, D., Knight, M., Emslie, R., Pani, M., Booth, V., Mahoney, S. & Buyanaa, C. (2017) The baby and the bathwater: trophy hunting, conservation and rural livelihoods. Unasylva, 68, 3-16.
Crosmary, W. G., Côté, S. D. & Fritz, H. (2015) The assessment of the role of trophy hunting in wildlife conservation. ANIMAL CONSERVATION, 18, 136-137.
David, M. (2016) Moving beyond the “Five Freedoms” by Updating the “Five Provisions” and Introducing Aligned “Animal Welfare Aims”. Animals, 6, 59.
Dubois, S. & Harshaw, H. W. (2013) Exploring "Humane" Dimensions of Wildlife. HUMAN DIMENSIONS OF WILDLIFE, 18, 1-19.
Encyclopedia, I. (2016) Completion Rate. Ipsos,.
Fox, M. W. (1986) In Advances in animal welfare science (eds M. W. Fox & L. D. Mickley).
Genovesi, P. & Bertolino, S. (2001) In The Great Reshuffling. Human Dimensions of Invasive Alien Species (ed J. A. McNeely).
HAMPTON, J. O. & TEH-WHITE, K. (2019) Animal Welfare, Social License, and Wildlife Use Industries. The Journal of Wildlife Management, 83, 12-21.
Knezevic, I. (2009) Hunting and Environmentalism: Conflict or Misperceptions. HUMAN DIMENSIONS OF WILDLIFE, 14, 12-20.
Lindsey, P., Balme, G., Funston, P., Henschel, P. & Hunter, L. T. B. (2016) Life after Cecil: Channelling global outrage into funding for conservation in Africa. Conservation Letters, 9, 296-301.
Mann, N. (2000) Dietary lean red meat and human evolution. European Journal of Nutrition volume, 39, 71-79.
Mann, N. (2013) Human evolution and diet: a modern conundrum of health versus meat consumption, or is it? Animal Production Science, 53, 1135-1142.
Mcmahon, C. R., Harcourt, R., Bateson, P. & Hindell, M. A. (2012) Animal welfare and decision making in wildlife research. BIOLOGICAL CONSERVATION, 153, 254-256.
Minin, E. D., Leader-Williams, N. & Bradshaw, C. J. A. (2016) Banning Trophy Hunting Will Exacerbate Biodiversity Loss. TRENDS IN ECOLOGY & EVOLUTION, 31, 99-102.
Morgera, E. & Wingard, J. (2009) Principles for developing sustainable wildlife management laws. Cic Technical,.
Muposhi, V. K., Edson, G., Paul, B. & M., M. S. (2016) Trophy Hunting, Conservation, and Rural Development in Zimbabwe: Issues, Options, and Implications. International Journal of Biodiversity, 2016, 1-16.
Ogutu, J. O., Hans-Peter, P., Said, M. Y., Ojwang, G. O., Njino, L. W., Kifugo, S. C., Wargute, P. W. & Rezende, P. S. (2016) Extreme Wildlife Declines and Concurrent Increase in Livestock Numbers in Kenya: What Are the Causes? PLoS One, 11, e163249.
Oommen, M. A., Cooney, R., Ramesh, M., Archer, M., Brockington, D., Buscher, B., Fletcher, R., Natusch, D. J. D., Vanak, A. T., Webb, G. & Shanker, K. (2019) The fatal flaws of compassionate conservation. CONSERVATION BIOLOGY,.
Paquet, P. & Darimont, C. (2010) Wildlife conservation and animal welfare: Two sides of the same coin? ANIMAL WELFARE, 19, 177-190.
Pereira, P. M. D. C. & Vicente, A. F. D. R. (2013) Meat nutritional composition and nutritive role in the human diet. MEAT SCIENCE, 93, 586-592.
SSC, I. (2012) IUCN SSC Guiding principles on trophy hunting as a tool for creating conservation incentives. Ver. 1.0. IUCN, Gland..
Trull & L., F. (2015) PETA undermines science and scientists. SCIENCE, 347, 834.
Wanger, T. C., Traill, L. W., Cooney, R., Rhodes, J. R. & Tscharntke, T. (2017) Trophy hunting certification. Nature Ecology & Evolution, 1, 1791-1793.
Williams, A. C. & Hill, L. J. (2017) Meat and Nicotinamide: A Causal Role in Human Evolution, History, and Demographics. International Journal of Tryptophan Research, 10, 1-23.
